# An Attention-based Predictive Agent for Handwritten Numeral/Alphabet Recognition via Generation

**Bonny Banerjee**                                           BBNERJEE@MEMPHIS.EDU
**Murchana Baruah**                                   MURCHANABARUAH@GMAIL.COM
*Institute for Intelligent Systems, and Department of Electrical & Computer Engineering, University of Memphis, Memphis, TN 38152, USA*

**Editor:** Editor's name

## Abstract

A number of attention-based models for either classification or generation of handwritten numerals/alphabets have been reported in the literature. However, generation and classification are done jointly in very few end-to-end models. We propose a predictive agent model that actively samples its visual environment via a sequence of glimpses. The attention is driven by the agent's sensory prediction (or generation) error. At each sampling instant, the model predicts the observation class and completes the partial sequence observed till that instant. It learns where and what to sample by jointly minimizing the classification and generation errors. Three variants of this model are evaluated for handwriting generation and recognition on images of handwritten numerals and alphabets from benchmark datasets. We show that the proposed model is more efficient in handwritten numeral/alphabet recognition than human participants in a recently published study as well as a highly-cited attention-based reinforcement model. This is the first known attention-based agent to interact with and learn end-to-end from images for recognition via generation, with high degree of accuracy and efficiency.

**Keywords:** Visual attention, glimpses, perception, proprioception, multimodal, handwritten numeral/alphabet recognition and generation.

## 1. Introduction

Perception and action are inextricably tied together as, in the real world, efficiency is as important as accuracy. Nature has evolved the visual system such that, to minimize resources, it learns to selectively attend to a few locations that provide information for the task at hand. This motivates our exploration of predictive agent models that observe the visual environment via a sequence of glimpses. Such agents predict, learn and act by minimizing sensory prediction error in a closed loop.

A number of works have explored attention-based agents that learn to sequentially sample their environment for spatial and spatiotemporal data generation. In this paper, we propose an attention-based predictive agent for handwritten numeral and alphabet recognition in images. The attention (action) is driven by the agent's sensory prediction error.

Attention-based models can be hard or soft (Xu et al., 2015; Elsayed et al., 2019). Hard-attention models make decisions by processing a part of the data, sampled via a sequence of glimpses. These models can be reinforcement-based (e.g., (Elsayed et al., 2019; Mnih et al., 2014)), unsupervised (e.g., (Gregor et al., 2015; Eslami et al., 2016)) or supervised (e.g., (Zheng et al., 2015)). Soft-attention models process the entire data but weigh the

features. Supervised (e.g., (Fukui et al., 2019)) and unsupervised (e.g., (Sang et al., 2020)) variants of these models have been reported. We propose a supervised (with class labels) hard-attention model that does not use any reinforcement.

Numerous attention-based models for either classification (e.g., (Mnih et al., 2014)) or generation (e.g., (Gregor et al., 2015; Baruah et al., 2022)) of handwritten numerals/alphabets have been reported in the literature. However, generation and classification are done jointly in very few end-to-end models. Two models deserve mention: semi-supervised learning with generative models proposed in (Kingma et al., 2014), and a multimodal variational autoencoder robust to missing data introduced in (Wu and Goodman, 2018). Though both models perform generation and classification of handwritten numerals (MNIST), only classification accuracy is reported in (Kingma et al., 2014) while only generation accuracy is reported in (Wu and Goodman, 2018). Further, none of them incorporate attention, i.e. an image is not sampled as a sequence of observations but presented in its entirety.

**Contributions.** In this paper, we propose an attention-based agent model that learns to classify handwritten numerals/alphabets from images by generating them. The novelty of this work is as follows:

- The proposed model implements a perception-action loop to optimize an objective function. *The action (attention) is modeled as proprioception in a multimodal setting* and is guided by perceptual prediction error, not by reinforcement. This kind of agent model was first introduced in (Baruah and Banerjee, 2020b), and has since been used to learn handwriting generation from images and videos (Baruah et al., 2022), human interaction generation (Baruah and Banerjee, 2020a), human interaction recognition via generation (Baruah et al., 2023a), and speech emotion recognition via generation (Baruah and Banerjee, 2022), but not for handwriting recognition. Also, no study has evaluated such a model in comparison to human efficiency.
- At each sampling instant, the model simultaneously classifies and completes the partial sequence of observations. Pattern completion allows prediction error computation which decides the next sampling location. Thus, attention emerges in our model and does not require learning feature weights.
- In the model, the pattern completion function maps the partial sequences of perceptual and proprioceptive observations to the class label and completed perceptual pattern. Three variants of this function are proposed. Their accuracies correlate with the number of trainable parameters.
- The model is more efficient than the human participants in a recently published study (Baruah et al., 2023b). On average, the study participants required 4.2, 4.7 and 4.9 samples to recognize a numeral, uppercase and lowercase alphabet respectively. When exposed to the same stimuli and conditions as the participants, our model requires 2.0, 4.5, 4.2 samples respectively. In contrast, a highly-cited attention-based reinforcement model (Mnih et al., 2014) falls short of human performance.

The rest of the paper is organized as follows. The proposed agent model is described in Section 2 and evaluated on various benchmark datasets in Section 3. The paper ends with concluding remarks in Section 4.

## 2. Models and Methods

### 2.1. Preliminaries

**Agent.** Anything that perceives from and acts upon its environment using sensors and actuators respectively is called an agent (Russell and Norvig, 2020).

**Perception** is the mechanism of interpreting sensory signals from the external environment by an agent (Han et al., 2016).

**Proprioception** is a form of perception in which the agent's environment is its own body (Baruah and Banerjee, 2020b). Internal perception of position, movement, and motion of body parts is due to proprioception (Han et al., 2016).

**Generative model.** Given a set of data points $x$, a generative model $p_{model}$ with parameters $\theta$ maximizes the log-likelihood, $\mathcal{L}(x; \theta)$, of the data.

**Evidence lower bound (ELBO).** Let the data $x$ be generated by a latent continuous random variable $z$. Then, computing the log-likelihood requires integrating the marginal likelihood, $\int p_{model}(x, z)dz$, which is intractable (Kingma and Welling, 2013). In variational inference, an approximation of the intractable posterior is optimized by defining an evidence lower bound (ELBO) on the log-likelihood, $\mathcal{L}(x; \theta) \leq \log p_{model}(x; \theta)$.

**Variational autoencoder (VAE)** is a multilayered generative model. It assumes an isotropic Gaussian prior, $p_\theta(z)$, and i.i.d. data samples. VAE maximizes the following ELBO (Kingma and Welling, 2013):

$$\mathbb{E}_{q_\phi(z|x)}[\log p_\theta(x|z)] - D_{\text{KL}}[q_\phi(z|x), p_\theta(z)] \tag{1}$$

where $p_\theta(x|z)$ and $q_\phi(z|x)$ are generative and recognition models respectively, $\mathbb{E}$ denotes expectation, and $D_{\text{KL}}$ denotes Kullback-Leibler divergence. The first and second terms capture accuracy and complexity respectively. The negative of this ELBO is also known as *variational free energy*, minimization of which has been hypothesized as a general principle guiding brain function (Friston, 2010).

**Saliency** lies in the eyes of an agent. Saliency of a location in an environment is a function of its neighborhood and an agent's internal model (Spratling, 2012; Friston et al., 2009).

### 2.2. Problem Statement

Let an environment in $m$ modalities be represented by a set of observable variables $\mathbf{X} = \{\mathbf{X}^{(1)}, \mathbf{X}^{(2)}, \ldots, \mathbf{X}^{(m)}\}$. The variable representing the $i$-th modality is a sequence: $\mathbf{X}^{(i)} = \langle X_1^{(i)}, X_2^{(i)}, \ldots, X_T^{(i)} \rangle$, where $T$ is the sequence length. Let $\mathbf{x}_{\leq t} = \{\mathbf{x}^{(1)}, \mathbf{x}^{(2)}, \ldots, \mathbf{x}^{(m)}\}$ be a partial observation of $\mathbf{X}$ such that $\mathbf{x}^{(i)} = \langle x_1^{(i)}, \ldots, x_t^{(i)} \rangle$, $1 \leq t \leq T$. Let $y$ represent the class label.

We define *pattern completion and classification* as the problem of accurately generating $\mathbf{X}$ and $y$ from the partial observation $\mathbf{x}_{\leq t}$. Given $\mathbf{x}_{\leq t}$ and a generative model $p_\theta$ with parameters $\theta$ and latent variables $z_{\leq t}$, the objective for pattern completion and classification at any time $t$ is to maximize the joint log-likelihood of $\mathbf{X}$ and $y$, i.e., $\arg\max_\theta \int log(p_\theta(\mathbf{X}, y|\mathbf{x}_{\leq t}, z_{\leq t}; \theta)p_\theta(z_{\leq t}))dz$.

### 2.3. Models

We solve the problem in three distinct ways as follows.

**Model M1** (ref. Fig. 2($a$)): The completed pattern and class label are generated from the latent variables. Mathematically, $\arg\max_{\theta} \int log(p_\theta(\mathbf{X}|\mathbf{x}_{\leq t}, z_{\leq t}; \theta)p_\theta(z_{\leq t}))dz + \arg\max_{\theta} \int log(p_\theta(y|\mathbf{x}_{\leq t}, z_{\leq t}; \theta)p_\theta(z_{\leq t}))dz$. The model is trained end-to-end.

**Model M2** (ref. Fig. 2($b$)): The class label is inferred from the partial observation. The latent variables are inferred from the class label and partial observation, as in (Kingma et al., 2014). Mathematically, $\arg\max_{\theta} \int log(p_\theta(\mathbf{X}|\mathbf{x}_{\leq t}, z_{\leq t}; \theta)p_\theta(z_{\leq t}))dz + \arg\max_{\phi} \log q_\phi(y_t|\mathbf{x}_{\leq t})$, where $q_\phi$ is a recognition model. The model is trained end-to-end.

**Model M3** (ref. Fig. 2($c$)): The class label is inferred from the completed pattern which is generated from the latent variables. The pattern completion model is trained first, $\arg\max_{\theta} \int log(p_\theta(\mathbf{X}|\mathbf{x}_{\leq t}, z_{\leq t}; \theta)p_\theta(z_{\leq t}))dz$. Then the classification model is trained, $\arg\max_{\pi} log(p_\pi(y|\mathbf{X}))$.

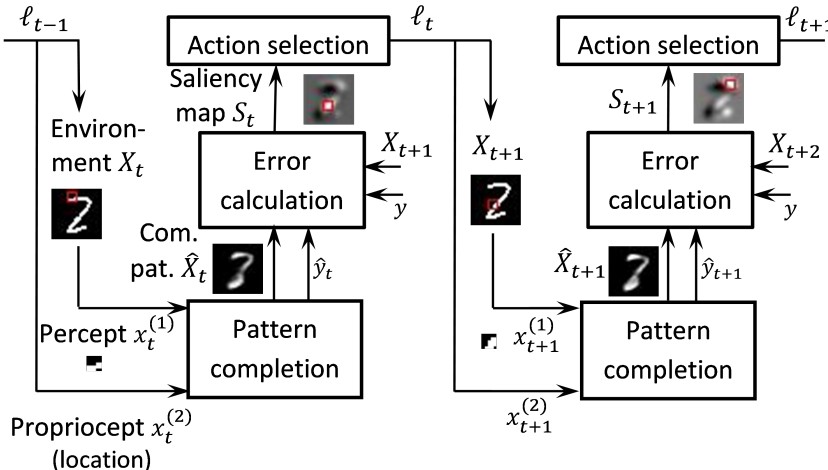

Figure 1: Different components of the proposed agent. Implementation of the pattern completion block is shown in Fig. 2.

Table 1: Variable dimensions as used in this paper. Here $(.)^{(1)}$, $(.)^{(2)}$ refer to visual perception and visual proprioception respectively; $T$ is maximum number of glimpses, $t$ is glimpse index or time, $n \times n$ is patch size, $M \times M$ is image size.

| $x_t^{(1)}$ | $x_t^{(2)}$ | $X_t$ | $S_t$ |
|---|---|---|---|
| $\{0,1\}^{n \times n}$ | $\mathbb{R}^2$ | $\{0,1\}^{M \times M}$ | $\mathbb{R}^{M \times M}$ |

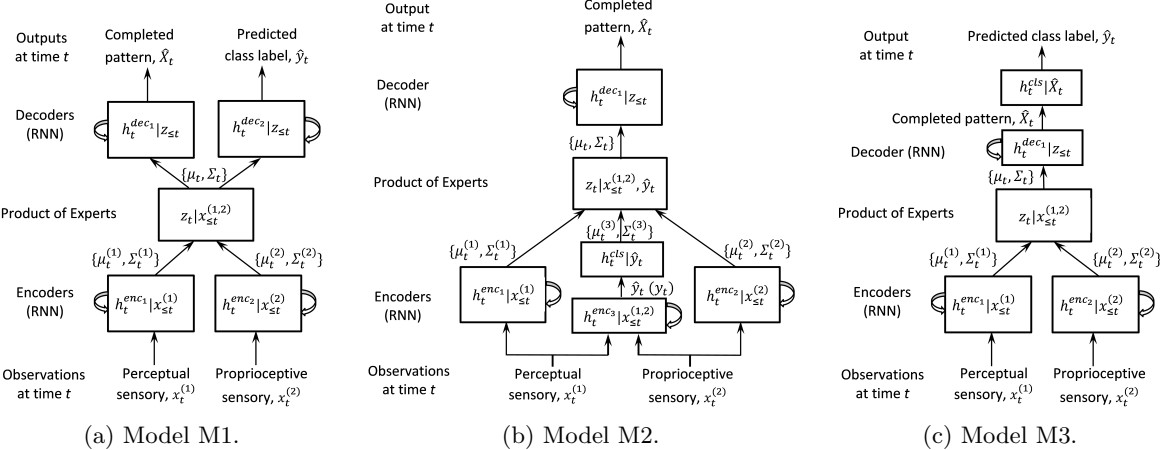

Figure 2: Three variants for implementing the pattern completion block in Fig. 1.

## 2.4. Agent Architecture

As shown in the block diagram in Fig. 1, environment, observation, pattern completion and classification, action selection and learning are the five components of the proposed agent architecture.

**1. Environment.** The environment is the source of sensory data. We consider a static environment (images) in this work.

**2. Observation.** Our agent sequentially samples its environment in two modalities: visual perception and visual proprioception. The 2D coordinates of the fixation location in the environment constitutes the proprioceptive observation while the visual stimuli at that location constitutes the corresponding perceptual observation, as in (Friston et al., 2012). See Table 1 for variable dimensions.

**3. Pattern completion.** At each sampling instant, the partial observation till that instant is completed using a multimodal variational recurrent neural network (MVRNN). Recognition and generation are the two processes involved in the operation of a MVRNN.

*Recognition (Encoder).* The recognition model, $q_\phi(z_t|\mathbf{x}_{\leq t})$ for M1 and M3, and $q_\phi(z_t|\mathbf{x}_{\leq t}, y_t)$ for M2, is a probabilistic encoder (Kingma and Welling, 2013). It produces a Gaussian distribution over the possible values of the code $z_t$ from which the given observations could have been generated.
**Model M1:** Two RNNs, each with one layer of long short-term memory (LSTM) units, constitute the recognition model. Each RNN infers the parameters of the approximate posterior distribution for each modality.
**Model M2:** In addition to the perceptual and proprioceptive modalities, the class label is an input modality. A fully-connected layer maps the class labels (inferred label $\hat{y}$ or given label $y$) to the parameters ($\mu^{(3)}$, $\Sigma^{(3)}$) of the approximate posterior density for the class label modality (ref. Fig. 2(b)).
**Model M3:** Same as M1.

The parameters for all modalities are combined using product of experts (PoE) (Wu and Goodman, 2018) to generate the joint distribution for the approximate posterior, $q_\phi(z_t|\mathbf{x}_{\leq t})$ for M1 and M3, and $q_\phi(z_t|\mathbf{x}_{\leq t}, y_t)$ for M2.

The prior can be sampled from a standard normal distribution $p_\theta(z_t) \sim \mathcal{N}(0, 1)$, as in (Gregor et al., 2015). The function of the encoder is shown in Lines 1–3 of Algorithm 2 and Lines 3–9 of Algorithm 3 (see Appendix A), where $RNN_\phi^{enc}$ represents the function of a LSTM unit, $\varphi^{enc}$ is a function that returns the mean and the logarithm of the standard deviation as a linear function of the hidden state, as in (Chung et al., 2015).

*Generation (Decoder).*

**Model M1:** The model, $p_\theta(X_t, y_t|\mathbf{x}_{\leq t}, z_{\leq t})$, generates the perceptual data and the class label from the latent variables, $z_t$, at each time step. The generative model consists of two RNNs, each with one layer of hidden LSTM units.

**Model M2:** The model, $p_\theta(X_t|\mathbf{x}_{\leq t}, z_{\leq t})$, generates the perceptual data from the latent variables, $z_t$. The generative model consists of one RNN with a single layer of hidden LSTM units.

**Model M3:** Same as M2.

Each RNN generates the parameters of the data distribution for a modality. The data is sampled from this distribution which can be multivariate Gaussian or Bernoulli. In our model, both $X_t$ and $y_t$ are sampled from a multivariate Bernoulli distribution with means inferred by the corresponding decoder RNN. In order to generate the perceptual data at any time step, the output from the perceptual RNN at the previous time step is added to the current perceptual RNN output before applying the sigmoid function, as in (Gregor et al., 2015). The decoder equations are shown in Lines 5–8 of Algorithm 2 and Lines 11–12 of Algorithm 3 (see Appendix A), where the function $RNN_\theta^{dec}$ is the same as $RNN_\phi^{enc}$.

**4. Classification.**

**Model M1:** The decoder infers the class label as a separate modality for each time step (ref. M1 in Generation (Decoder)).

**Model M2:** The class labels are inferred from the partial observations, $\mathbf{x}_{\leq t}$, at every time step. An RNN with LSTM units is used as a hidden layer, along with a softmax classifier. The function of the classifier is shown in Lines 1–2 of Algorithm 3 (see Appendix A).

**Model M3:** A classifier[1] is trained separately to infer the class labels from the perceptual data. During training, the input to the classifier is the true perceptual data. During testing, the input is the predicted perceptual data.

**5. Action selection.** In our model, action selection is to decide the location in the environment to sample from. At any time $t$, a saliency map $S_t$ is computed which assigns a salience score $S_t^{(\ell)}$ to each location $\ell$.

$$S_t^{(\ell)} = D_{KL}(p(X_{t+1,\ell})||p_\theta(X_{t+1,\ell}|z_{\leq t}, \mathbf{x}_{\leq t})) \tag{2}$$

where $p(X_{t+1,\ell})$ is the true data distribution at location $\ell$ and is sampled from a Bernoulli distribution. KL divergence, also known as *relative entropy*, is a measure of information gain achieved by using the true distribution, $p(X_{t+1,\ell})$, instead of the predicted distribution, $p_\theta(X_{t+1,\ell}|z_{\leq t}, \mathbf{x}_{\leq t})$. Thus, the saliency map is a function of the prediction error. The most salient location is computed from this saliency map which constitutes the sampling location.

---

1. We used a CNN classifier with code borrowed from https://chromium.googlesource.com/external/github.com/tensorflow/tensorflow/+/r0.10/tensorflow/g3doc/tutorials/mnist/pros/index.md.

The saliency map is smoothed using a Gaussian kernel $\mathcal{N}(.,\sigma)$. The sampling location is chosen as:

$$\ell_t = \operatorname*{argmax}_{\ell_t \in \{1,2,\ldots,M^2\}} \operatorname{conv}(\mathcal{N}(.,\sigma), S_t) \tag{3}$$

where $\sigma = 2$. Each sample is a $n \times n$ patch centered at $\ell_t$.

The salient location $\ell_t$ at any time $t$ is the proprioceptive observation $x_{t+1}^{(2)}$ for time $t+1$. Hence, prediction error (saliency) guides the sampling of a scene in our model. Unlike typical multimodal models, the two modalities in our model interact at the observation level as the perceptual prediction error provides the observation for the visual proprioceptive modality. The most salient location is the location that yields the maximum information gain in the environment. These are the locations where the agent's prediction error is the highest given all the past observations. The agent attends to these locations to update its internal model.

**6. Learning.** The objective is to maximize Equ. 4, 5 and 6 for M1, M2 and M3 respectively. It can be derived from the objectives for multimodal VAE (Wu and Goodman, 2018), variational RNN (Chung et al., 2015) and VAE for classification (Kingma et al., 2014). See Appendix A for derivations.

$$\mathbb{E}_{q_\phi(z_{\leq T}|\mathbf{x}_{\leq T})}\Big[\sum_{t=1}^{T}\lambda_1 \log p_\theta(X_t|z_{\leq t},\mathbf{x}_{\leq t}) + \lambda_2 \log p_\theta(y_t|z_{\leq t},\mathbf{x}_{\leq t})\Big] - \sum_{t=1}^{T}\beta D_{KL}\big(q_\phi(z_t|\mathbf{x}_{\leq t}), p_\theta(z_t)\big) \tag{4}$$

where $\lambda_1$, $\lambda_2$, $\beta$ are the weights balancing the terms.

$$\mathbb{E}_{q_\phi(z_{\leq T}|\mathbf{x}_{\leq T},y_{\leq T})}\Big[\sum_{t=1}^{T}\log p_\theta(X_t|z_{\leq t},\mathbf{x}_{\leq t}) + \log p_\theta(y_t)\Big] - \sum_{t=1}^{T}D_{KL}\big(q_\phi(z_t|\mathbf{x}_{\leq t},y_t), p_\theta(z_t)\big)$$
$$+ \sum_{t=1}^{T}\alpha \log q_\phi(y_t|\mathbf{x}_{\leq t}) \tag{5}$$

where $\alpha$ controls the relative weight between generative and purely discriminative learning.

$$\mathbb{E}_{q_\phi(z_{\leq T}|\mathbf{x}_{\leq T})}\Big[\sum_{t=1}^{T}\log p_\theta(X_t|z_{\leq t},\mathbf{x}_{\leq t})\Big] - \sum_{t=1}^{T}D_{KL}\big(q_\phi(z_t|\mathbf{x}_{\leq t}), p_\theta(z_t)\big) + \log q_\pi(y|X) \tag{6}$$

where $q_\pi(y|X)$ is the classification model whose input is the entire image (completed pattern) and not a sequence of observations. Hence the subscript $t$ is dropped.

We assume a one-to-one mapping between the agent's body and its environment, i.e. between the oculomotor muscles to the locations in the image. This assumption allows us to map from the perceptual space $\ell$ to the proprioceptive space $x^{(2)}$ using a simple function $g_3$ (ref. Line 9 of Algorithm 1).

## 2.5. Metrics for comparing fixation maps

In order to evaluate the action mechanism of our model, we compare the fixation map obtained from the sequence of locations sampled by our model with that of the fixation map obtained from participants' data in (Baruah et al., 2023b). The fixation map is computed by

Table 2: Evaluation of fixation maps from RAM and our model (Model 1) for the stimuli presented in the MTurk experiments, averaged over all classes and samplings. Standard deviations are included in parenthesis.

| Metric | MNIST | | EMNIST uppercase | | EMNIST lowercase | |
|--------|-------|-----|------------------|-----|------------------|-----|
| | Our model (M1) | RAM | Our model (M1) | RAM | Our model (M1) | RAM |
| KL | 22.44(7.50) | 22.50(7.48) | 22.90(7.55) | 22.96(7.24) | 22.30(7.37) | 22.23(7.16) |
| CC | 0.02(0.01) | 0.01(0.00) | 0.02(0.01) | 0.01(0.00) | 0.02(0.01) | 0.01(0.00) |
| SIM | 0.18(0.11) | 0.17(0.09) | 0.16(0.10) | 0.16(0.07) | 0.18(0.10) | 0.18(0.09) |

assigning each location a value equal to the frequency of its selection, and then normalizing the values to create a distribution over all locations.

For metrics comparing two fixation maps, $P$ and $Q$, we closely follow (Bylinskii et al., 2018). We use three distribution-based metrics: KL divergence (KL), Pearson correlation coefficient (CC), and Similarity (SIM), to compare the distribution of sampling locations from a model with that from the participants as recorded in the collected data.

**KL divergence.** (Bylinskii et al., 2018) Given two image distributions, $P$ and $Q$, the KL divergence $KL(P, Q)$ measures the loss of information when $Q$ is used to approximate $P$. This is calculated for each pixel $k$ as: $KL(P_k, Q_k) = P_k \log \left( \epsilon + \frac{P_k}{Q_k + \epsilon} \right)$, where $\epsilon$ is a very small real number. Lower KL divergence for $k$ implies $P_k$ and $Q_k$ are similar. KL divergence is highly sensitive to zero values.

**CC** can evaluate the linear relationship between two maps as (Bylinskii et al., 2018): $CC(P, Q) = \frac{\sigma(P,Q)}{\sigma(P)\sigma(Q)}$, where $\sigma$ is the variance or covariance. Since CC is symmetric, it fails to infer whether differences between fixation maps are due to false positives or false negatives.

**SIM** is measured as (Bylinskii et al., 2018): $SIM(P, Q) = \sum_k \min(P_k, Q_k)$, where $\sum_k P_k = \sum_k Q_k = 1$. Like CC, SIM is symmetric and inherits the same drawback. Also, SIM is very sensitive to missing values, and penalizes predictions that fail to account for the ground truth density.

These metrics do not compare the sequence of fixations. This is inconsequential in the current work because recognizing a numeral or alphabet does not require sampling the image in a particular order. Baruah et al. (2022) have shown that a predictive agent saccades when exposed to images of handwritten numerals or alphabets, and tracks when exposed to videos of the formation of the same handwritten numerals. In both cases, the agent learns to complete the proprioceptive pattern or the sequence of expected salient (or sampling) locations. See Fig. 2 in (Baruah et al., 2022). The agent model in the current work also learns to complete the proprioceptive pattern in the same way, though this is not shown.

## 3. Experimental Results

### 3.1. Datasets

Our model is evaluated using the following datasets:
(1) MNIST (LeCun et al., 1998) is a dataset of handwritten numerals $\{0, 1, \ldots, 9\}$, consisting of 60,000 training and 10,000 test images ($28 \times 28$ pixels).

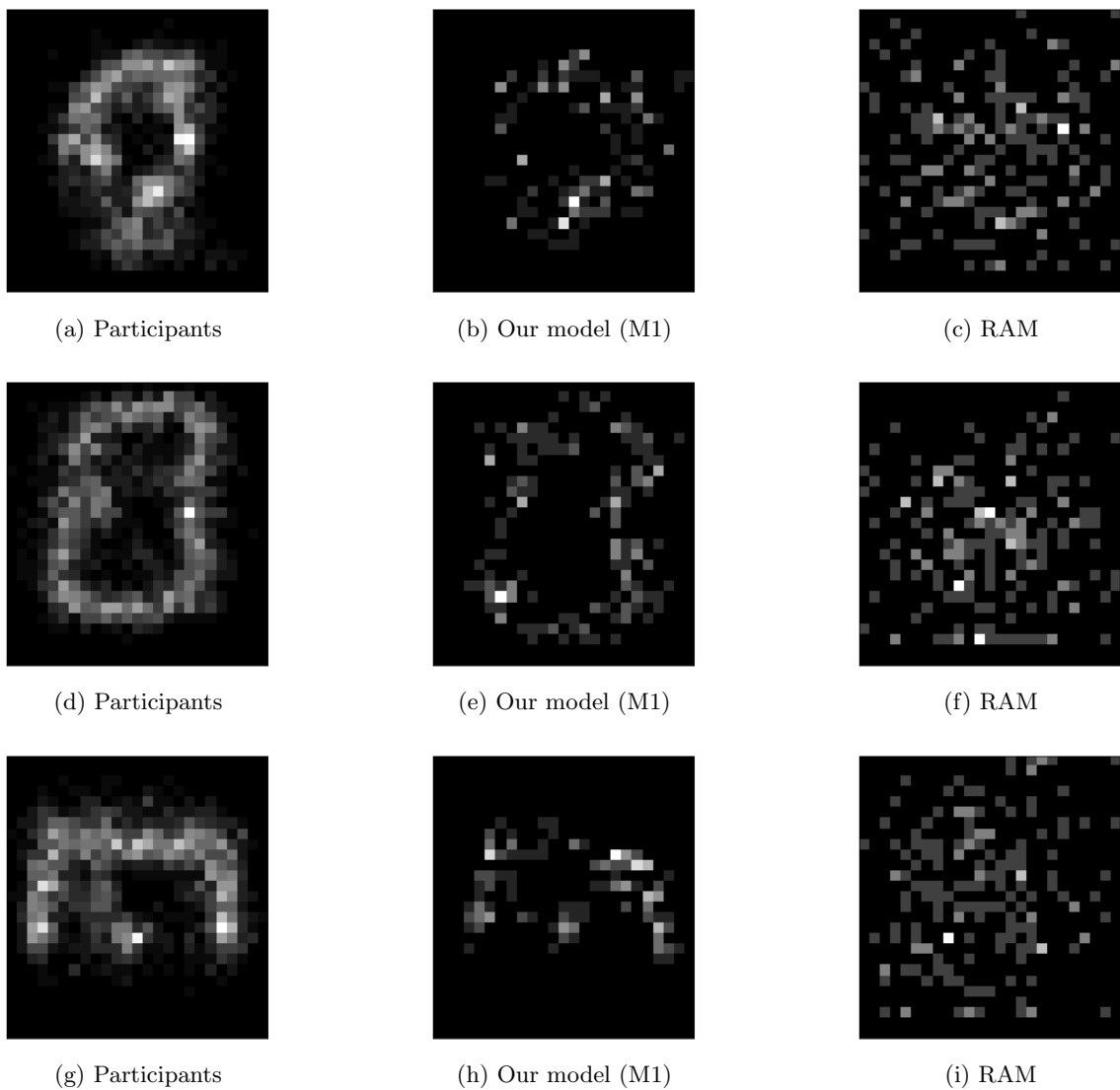

Figure 3: Comparison of the distribution of the sequence of fixations over a class for different cases; classes '9', 'B', 'm' are shown in rows 1 to 3 respectively. The fixations are scattered in case of RAM, our model shows similar pattern with the participants data.

(2) EMNIST (Cohen et al., 2017) is a balanced dataset of handwritten English alphabets in uppercase and lowercase, consisting of 124,800 training and 20,800 test images ($28 \times 28$ pixels).
(3) AttentionMNIST (Baruah et al., 2023b) is a dataset[2] consisting of a sequence of time-stamped samples from MNIST and EMNIST datasets, collected from human participants using MTurk. Each sample consists of: (1) the location in the image selected by the participant, (2) the class(es) selected by the participant, and (3) the time taken by the

---

2. We downloaded the AttentionMNIST dataset from https://github.com/Murchana/AttentionMNIST.

participant to register the current sample (i.e. the time elapsed between registering the last and current samples). The total time allowed to each participant for sampling $T = 12$ locations of an image is limited to six minutes. This data is recorded from 15 distinct stimuli from each class for MNIST, EMNIST uppercase, and EMNIST lowercase letters. The dataset is collected from 382 distinct participants. It consists of 1736 samples from MNIST, 4431 samples from EMNIST uppercase, and 4315 samples from EMNIST lowercase, and 169.1 responses per class on average.

## 3.2. Experimental setup

The generative, recognition and classification models consist of 512, 128, 128 hidden units respectively. The latent variable dimension is 20. These parameters are estimated experimentally, and are consistent with model parameters reported in the literature. For example, the multimodal model in (Wu and Goodman, 2018) uses latent variable dimension of 64 and two MLP hidden layers of 512 units each for MNIST generation and classification, the model in (Gregor et al., 2015) uses latent variable dimension of 100 and an RNN hidden layer of 256 units for MNIST generation, and the model in (Mnih et al., 2014) uses an RNN hidden layer of 256 units for MNIST classification.

Maximum number of glimpses $T = 12$, and minibatch size is 100. The parameters $\beta$, $\lambda_1$, are fixed to 1, $\lambda_2$ and $\alpha$ are fixed to 5000. The model is learned end-to-end using backpropagation and Adam optimization (Kingma and Ba, 2014) with a learning rate of $10^{-3}$. These hyperparameters are estimated via cross-validation using 10,000 images from the training set. The first observation is sampled from the center pixel of an image, as in the participants' data (Baruah et al., 2023b).

We use a dropout probability of 0.7 to prevent overfitting. The dropout is applied at the decoder hidden layers for all the modalities in M1 and M3, and both the decoder hidden layer and the classification hidden layer for M2. Additionally, the KL divergence term in the objective function also acts as a regularizer (Kingma and Welling, 2013) that prevents overfitting.

### 3.2.1. EVALUATION

The quality of the generated images is evaluated using negative log-likelihood (NLL), as in (Gregor et al., 2015), and the class prediction is evaluated by classification accuracy. The three metrics, KL, CC and SIM, are used to evaluate the fixation maps obtained from the sequence of sampled locations. The efficiency of the model is evaluated by the number of glimpses required for accurate prediction, on the sampled MNIST and EMNIST datasets (Baruah et al., 2023b).

As in (Baruah et al., 2023b), we compare the efficiency and fixation maps with a highly-cited reinforcement model, recurrent attention model (RAM) (Mnih et al., 2014), that reports experimental results on the MNIST dataset. RAM classifies images using a sequence of glimpses. The next location is chosen stochastically from a distribution parameterized by a location network. For a fair comparison with the participants, in RAM[3], we fixed the sequence length at $T = 12$, the first sampling location at the image center, the input

---

3. We use the RAM implementation from github.com/hehefan/Recurrent-Attention-Model.

Table 3: Classification accuracy and NLL on the test set reported after the final glimpse.

| Dataset | Variants of the proposed model | Accuracy (%) | NLL ($\leq$) |
|---------|-------------------------------|--------------|--------------|
| MNIST | M1 | 96.3 | 76.5 |
| | M2 | 92.3 | 107.0 |
| | M3 (pretrained) | 94.6 | 76.1 |
| | M4 (not end-to-end) | 82.9 | 76.1 |
| EMNIST | M1 | 90.2 | 125.8 |
| | M2 | 80.4 | 82.6 |
| | M3 (pretrained) | 88.5 | 78.9 |
| | M4 (not end-to-end) | 75.4 | 78.9 |

Table 4: Classification accuracy and NLL on the stimuli presented to the participants in (Baruah et al., 2023b), reported after the final glimpse.

| Dataset | Variants of the proposed model | Accuracy (%) | NLL ($\leq$) |
|---------|-------------------------------|--------------|--------------|
| MNIST | M1 | 100 | 71.3 |
| | M2 | 96 | 102.5 |
| | M3 (pretrained) | 98.7 | 71.8 |
| | M4 (not end-to-end) | 20.7 | 71.8 |
| EMNIST upp. | M1 | 98.7 | 129.7 |
| | M2 | 90.2 | 91.7 |
| | M3 (pretrained) | 98.7 | 83.9 |
| | M4 (not end-to-end) | 76.9 | 83.9 |
| EMNIST low. | M1 | 95.6 | 111.0 |
| | M2 | 85.4 | 66.8 |
| | M3 (pretrained) | 96.9 | 62.3 |
| | M4 (not end-to-end) | 74.9 | 62.3 |

observation to a $5 \times 5$ patch with the selected location as its center, and modified the reward function according to the experimental setup in (Baruah et al., 2023b).

In addition to the three variants (M1, M2, M3), we include one more variation of our model in which the generative model is trained as in M3, and then an RNN with LSTM units is used to classify the data from the latent variables. We refer to this as **Model M4**. Unless otherwise stated, "our model" refers to M1 throughout the rest of the paper.

### 3.3. Evaluation results

#### 3.3.1. EVALUATION FOR ACCURACY

When both the classification and the pattern completion modality are trained end-to-end as in M1 and M2, NLL increases (ref. Tables 3, 4). As the model is trained to learn generation and classification tasks at the same time, the model is not able to perform well, due to which the accuracy in the generation modality lowers. When the pattern completion

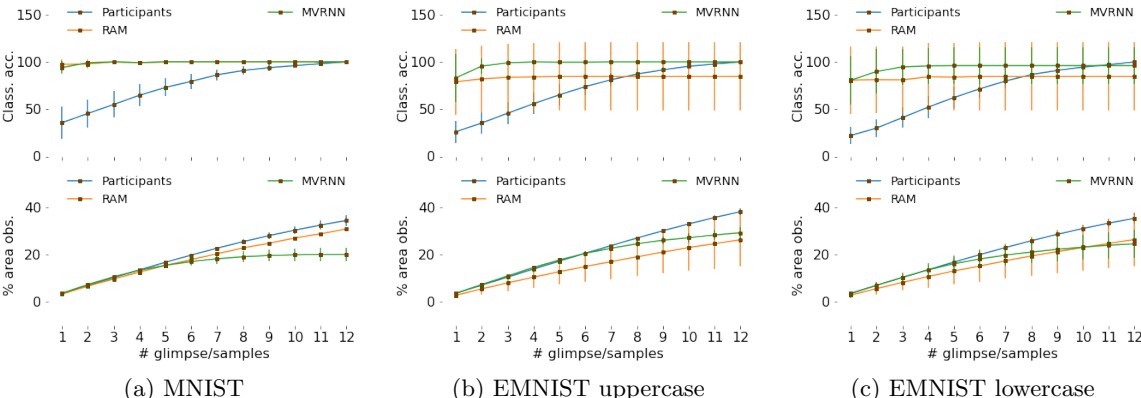

Figure 4: Errorbar plot showing the change in classification accuracy and percentage of image area observed by the participants in (Baruah et al., 2023b), RAM (Mnih et al., 2014) and our model (M1, MVRNN) with number of glimpses or samples.

and the classification modalities are trained separately, as the model is trained to learn the generation task only, the NLL is the lowest (ref. Tables 3, 4).

The classification accuracy from M1 is higher than M2 in all cases (ref. Tables 3, 4). In M1, the classification modality shares parameters with the generation modality, whereas in M2, the classification modality does not share parameters with the generation modality, though in both cases the generation modality shares parameters with the classification modality. Thus, the generation modality contributes more to the classification accuracy of M1 than of M2. The classification accuracy for M3 is very close to M1 and the classification accuracy for M4 is the lowest (ref. Tables 3, 4). M3 utilizes a CNN-based classifier; it yields higher classification accuracy than M4, which utilizes an RNN-based classifier.

### 3.3.2. Evaluation of fixation maps

Results from comparing the fixation maps from RAM and our model (M1) with the participants' data (Baruah et al., 2023b) are shown in Table 2. KL is higher due to its sensitivity to zero values. This implies several locations are sampled by the participants (as there are multiple participants for each stimulus) but not by RAM or our model. KL is lower for our model (M1) than RAM for most cases. SIM and CC are either higher for our model than RAM, or comparable for both the models.

Clearly, between our model (M1) and RAM, the fixation maps generated by the former are more similar to those generated by the participants. Visualization of the fixation maps in Figs. 3, 5, A1, A2 also shows that the maps obtained from our model are more similar to the participants'. As multiple participants responded to each stimulus, there are many more points for participants than for RAM or our model in the visualizations.

As our model is based on saliency computed using prediction error and the human brain is closely linked with predictive coding (Friston, 2010), this can possibly explain greater similarity of the fixation maps for our model. These experiments can be used as a baseline for evaluating locations sampled by an attention model.

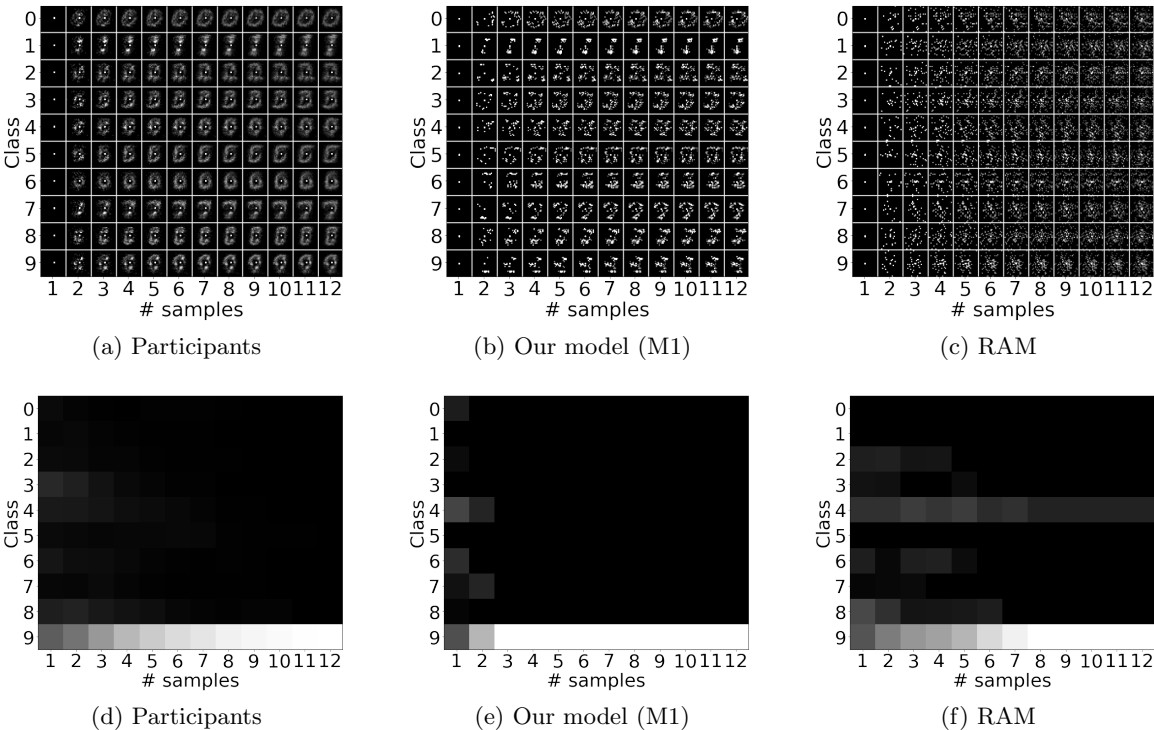

Figure 5: (a)–(c) Distribution of sampling locations (or fixation maps) for each numeral and each sampling instant. (d)–(f) Class distribution for class '9'. Qualitatively, the participants' fixation maps are more similar to our model's than RAM's. The distributions are obtained by averaging the responses over all stimuli presented from each class. Each row corresponds to a class, and each column corresponds to a sampling instant which increases from left to right. Also see Figs. A1 and A2 in Appendix B, which show similar results for uppercase and lowercase alphabets respectively.

The attention mechanism in our model differs from most models (including RAM) from behavioral and algorithmic perspectives. Typically, end-to-end attention-based models learn all parameters, including attention weights, by optimizing an objective function. In most of these models, attention is an internal mechanism that does not have a corresponding external behavior. The attention parameters play a role similar to any other parameter in the model. In our model, attention is a parameterless mechanism that emerges due to prediction error, which drives action/behavior (ref. Eqs. 2, 3). This mechanism is interpretable as the model simply attends to its unexpected observations.

### 3.3.3. Evaluation for efficiency

In (Baruah et al., 2023b), a participant can select multiple classes at any instant. For the proposed and RAM models, instead of predicting the highest probable class, we consider the mean probability over all the classes as a threshold and predict the set of classes with

probabilities greater than the threshold. We store the sampling or glimpse number after which the participant and the models select only the correct class.

The average number of samplings required by a participant to accurately predict a class is quite low. On average, it takes 4.2, 4.7, 4.9 samples for MNIST, EMNIST uppercase and lowercase images (Baruah et al., 2023b). RAM requires 3.7, 8.5, 7.6 samples to recognize MNIST numerals, uppercase and lowercase EMNIST alphabets respectively. Thus, in comparison to the participants, under the same experimental conditions, RAM is less efficient. Our model requires 2.0, 4.5, 4.2 samples to recognize MNIST numerals, uppercase and lowercase EMNIST alphabets respectively.

In order to yield the same accuracy, our model requires fewer glimpses than RAM and the participants (ref. Fig. 4). Hence, our model is more efficient. This is also validated by the class distribution plots shown in (d–f) of Figs. 5, A1, A2. We also observe that the classification accuracy over glimpses plots for RAM and our model are mostly flat (ref. Fig. 4). This is because, since we are using a threshold to select multiple classes from these models as stated above, the correct class is selected in most of the glimpses, which does not change the classification accuracy much over glimpses. The proportion of area observed increases with glimpses for RAM and the participants, but it saturates after a few glimpses for our model, particularly in Fig. 4(a). As there is no inhibition of return used in our model during sampling, the model is allowed to sample near the already sampled locations, which may have led to this pattern.

## 4. Conclusions

We proposed an attention-based agent model for handwritten numeral/alphabet recognition via a sequence of glimpses. The attention is driven by the agent's sensory prediction (or generation) error. At each sampling instant, the agent completes and classifies the partial sequence observed till that instant. End-to-end attention-based models that perform simultaneous generation and classification of handwritten numerals/alphabets is scarce. Our agent model is learned by jointly minimizing the classification and generation errors. Three variants of this model are evaluated on benchmark datasets. Their accuracies are comparable and correlate with the model size. Our experiments reveal that the proposed model is more data-efficient in handwritten numeral/alphabet recognition than human participants as well as a highly-cited attention-based reinforcement model, under the same conditions and stimuli. Qualitatively, the participants' fixation maps are more similar to our model's fixation maps than the reinforcement model's. To the best of our knowledge, this is the first attention-based end-to-end agent of its kind for recognition via generation, with high degree of accuracy and efficiency.

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

## Appendix A. Loss function derivation and pseudo code

### A.1. Model M1

Here we derive the objective function in Eq. 4. The generative and recognition models are factorized as:

$$p_\theta(X_{\leq T}, y_{\leq T}, z_{\leq T}|\mathbf{x}_{\leq T}) = \prod_{t=1}^{T} p_\theta(X_t, y_t|z_{\leq t}, \mathbf{x}_{\leq t})p_\theta(z_t)$$

$$q_\phi(z_{\leq T}|\mathbf{x}_{\leq T}) = \prod_{t=1}^{T} q_\phi(z_t|\mathbf{x}_{\leq t})$$

The variational lower bound (ELBO) on the joint log-likelihood of the generated data, $\log p_\theta(X_{\leq T}, y_{\leq T}|\mathbf{x}_{\leq T})$, is derived as:

$$\mathbb{E}_{q_\phi(z_{\leq T}|\mathbf{x}_{\leq T})}\Big[\log p_\theta(X_{\leq T}, y_{\leq T}|\mathbf{x}_{\leq T})\frac{q_\phi(z_{\leq T}|\mathbf{x}_{\leq T})}{q_\phi(z_{\leq T}|\mathbf{x}_{\leq T})}\Big]$$

$$= \mathbb{E}_{q_\phi(z_{\leq T}|\mathbf{x}_{\leq T})}\Big[\log \frac{p_\theta(X_{\leq T}, y_{\leq T}, z_{\leq T}|\mathbf{x}_{\leq T})}{p_\theta(z_{\leq T}|\mathbf{x}_{\leq T})}\frac{q_\phi(z_{\leq T}|\mathbf{x}_{\leq T})}{q_\phi(z_{\leq T}|\mathbf{x}_{\leq T})}\Big]$$

$$= \mathbb{E}_{q_\phi(z_{\leq T}|\mathbf{x}_{\leq T})}\Big[\sum_{t=1}^{T}\log \frac{p_\theta(X_t, y_t|z_{\leq t}, \mathbf{x}_{\leq t})p_\theta(z_t)}{p_\theta(z_t|\mathbf{x}_{\leq t})}\frac{q_\phi(z_t|\mathbf{x}_{\leq t})}{q_\phi(z_t|\mathbf{x}_{\leq t})}\Big]$$

$$= \mathbb{E}_{q_\phi(z_{\leq T}|\mathbf{x}_{\leq T})}\Big[\sum_{t=1}^{T}\Big[\log p_\theta(X_t, y_t|z_{\leq t}, \mathbf{x}_{\leq t}) - \log \frac{q_\phi(z_t|\mathbf{x}_{\leq t})}{p_\theta(z_t)} + \log \frac{q_\phi(z_t|\mathbf{x}_{\leq t})}{p_\theta(z_t|\mathbf{x}_{\leq t})}\Big]\Big]$$

$$\geq \mathbb{E}_{q_\phi(z_{\leq T}|\mathbf{x}_{\leq T})}\Big[\sum_{t=1}^{T}\log p_\theta(X_t, y_t|z_{\leq t}, \mathbf{x}_{\leq t})\Big] - \sum_{t=1}^{T} D_{KL}\big(q_\phi(z_t|\mathbf{x}_{\leq t}), p_\theta(z_t)\big)$$

We assume, the modalities $X_t$ and $y_t$ are conditionally independent given the common latent variables (Wu and Goodman, 2018) and all observations till the current time. Therefore,

$$\log p_\theta(X_{\leq T}, y_{\leq T}|\mathbf{x}_{\leq T}) \geq \mathbb{E}_{q_\phi(z_{\leq T}|\mathbf{x}_{\leq T})}\Big[\sum_{t=1}^{T}\lambda_1 \log p_\theta(X_t|z_{\leq t}, \mathbf{x}_{\leq t}) + \lambda_2 \log p_\theta(y_t|z_{\leq t}, \mathbf{x}_{\leq t})\Big]$$

$$- \sum_{t=1}^{T}\beta D_{KL}\big(q_\phi(z_t|\mathbf{x}_{\leq t}), p_\theta(z_t)\big) \qquad \text{(A1)}$$

where $\lambda_1$, $\lambda_2$, $\beta$ are the weights balancing the terms.

---

**Algorithm 1:** Learning the proposed network

---

Initialize parameters of the generative model $\theta$, recognition model $\phi$, sequence length $T$. Initialize optimizer parameters: $\beta_1 = 0.9$, $\beta_2 = 0.99$, $\eta = 0.001$, $\epsilon = 10^{-10}$. Initialize $x_1^{(1)} \leftarrow F(X_1, \ell_0)$, $x_1^{(2)} \leftarrow g_3(\ell_0)$, where $\ell_0$ is the initial sampling location (ref. Experimental setup in Section 3), $g_3$ is an identity function (ref. Action selection in Section 2.4), and the function $F$ extracts a sample $x^{(1)}$ (e.g., $5 \times 5$ patch) from the environment $X$ (e.g., $28 \times 28$ image) at location $\ell$ (e.g., center of the image).

**1** **while** *true* **do**
**2**    **for** $\tau \leftarrow 1$ *to* $T$ **do**
      **Model M1**
**3**       $\hat{X}_\tau, \hat{y}_\tau \leftarrow PatComClassM1(x_{1:\tau}^{(1:2)})$
      **Model M2**
**4**       $\hat{X}_\tau, \hat{y}_\tau \leftarrow PatComClassM2(x_{1:\tau}^{(1:2)})$
      **Model M3**
**5**       $\hat{X}_\tau \leftarrow PatComClassM1(x_{1:\tau}^{(1:2)})$
**6**       $\hat{y}_\tau \leftarrow Classifier(\hat{X}_\tau)$

      **Saliency Computation**
**7**       $S_\tau \leftarrow g_1(X_{\tau+1}, \hat{X}_\tau)$     [ref. Eq. 2]
**8**       $\ell_\tau \leftarrow g_2(S_\tau)$    [ref. Eq. 3]
**9**       $x_{\tau+1}^{(2)} \leftarrow g_3(\ell_\tau)$
**10**     $x_{\tau+1}^{(1)} \leftarrow F(X_{\tau+1}, \ell_\tau)$

      **Learning**
**11**     Update $\{\theta, \phi\}$ or $\{\theta, \phi, \pi\}$ by maximizing Eq. 4, 5 or 6.
   **end**
**end**

---

## A.2. Model M2

Here we derive the objective function in Eq. 5. The generative and recognition models are factorized as:

$$p_\theta(X_{\leq T}, y_{\leq T}, z_{\leq T}|\mathbf{x}_{\leq T}) = \prod_{t=1}^{T} p_\theta(X_t, y_t|z_{\leq t}, \mathbf{x}_{\leq t})p_\theta(z_t)$$

$$q_\phi(z_{\leq T}|\mathbf{x}_{\leq T}, y_{\leq T}) = \prod_{t=1}^{T} q_\phi(z_t|\mathbf{x}_{\leq t}, y_t)$$

The variational lower bound (ELBO) on the log-likelihood of the generated data, $\log p_\theta(X_{\leq T}, y_{\leq T}|\mathbf{x}_{\leq T})$, when the true label is given is derived as:

$$\mathbb{E}_{q_\phi(z_{\leq T}|\mathbf{x}_{\leq T}, y_{\leq T})}\left[\log p_\theta(X_{\leq T}, y_{\leq T}|\mathbf{x}_{\leq T})\frac{q_\phi(z_{\leq T}|\mathbf{x}_{\leq T}, y_{\leq T})}{q_\phi(z_{\leq T}|\mathbf{x}_{\leq T}, y_{\leq T})}\right]$$

$$= \mathbb{E}_{q_\phi(z_{\leq T}|\mathbf{x}_{\leq T}, y_{\leq T})}\left[\log \frac{p_\theta(X_{\leq T}, z_{\leq T}, y_{\leq T}|\mathbf{x}_{\leq T})}{p_\theta(z_{\leq T}|\mathbf{x}_{\leq T}, y_{\leq T})}\frac{q_\phi(z_{\leq T}|\mathbf{x}_{\leq T}, y_{\leq T})}{q_\phi(z_{\leq T}|\mathbf{x}_{\leq T}, y_{\leq T})}\right]$$

$$= \mathbb{E}_{q_\phi(z_{\leq T}|\mathbf{x}_{\leq T}, y_{\leq T})}\left[\sum_{t=1}^{T}\log \frac{p_\theta(X_t|z_{\leq t}, \mathbf{x}_{\leq t})p_\theta(z_t)p_\theta(y_t)}{p_\theta(z_t|\mathbf{x}_{\leq t}, y_t)}\frac{q_\phi(z_t|\mathbf{x}_{\leq t}, y_t)}{q_\phi(z_t|\mathbf{x}_{\leq t}, y_t)}\right]$$

$$= \mathbb{E}_{q_\phi(z_{\leq T}|\mathbf{x}_{\leq T}, y_{\leq T})}\left[\sum_{t=1}^{T}\left[\log p_\theta(X_t|z_{\leq t}, \mathbf{x}_{\leq t}) + \log p_\theta(y_t) - \log \frac{q_\phi(z_t|\mathbf{x}_{\leq t}, y_t)}{p_\theta(z_t)}\right.\right.$$

$$\left.\left. + \log \frac{q_\phi(z_t|\mathbf{x}_{\leq t}, y_t)}{p_\theta(z_t|\mathbf{x}_{\leq t}, y_t)}\right]\right]$$

$$\geq \mathbb{E}_{q_\phi(z_{\leq T}|\mathbf{x}_{\leq T}, y_{\leq T})}\left[\sum_{t=1}^{T}\log p_\theta(X_t|z_{\leq t}, \mathbf{x}_{\leq t}) + \log p_\theta(y_t)\right] - \sum_{t=1}^{T}D_{KL}\big(q_\phi(z_t|\mathbf{x}_{\leq t}, y_t), p_\theta(z_t)\big)$$

$$= \mathbb{E}_{q_\phi(z_{\leq T}|\mathbf{x}_{\leq T}, y_{\leq T})}\left[\sum_{t=1}^{T}(\log p_\theta(X_t|z_{\leq t}, \mathbf{x}_{\leq t}) + \log p_\theta(y_t))\right] - \sum_{t=1}^{T}D_{KL}\big(q_\phi(z_t|\mathbf{x}_{\leq t}, y_t), p_\theta(z_t)\big)$$

After adding the classification loss, the final objective function can be written as:

$$\mathbb{E}_{q_\phi(z_{\leq T}|\mathbf{x}_{\leq T}, y_{\leq T})}\left[\sum_{t=1}^{T}\log p_\theta(X_t|z_{\leq t}, \mathbf{x}_{\leq t}) + \log p_\theta(y_t)\right]$$

$$- \sum_{t=1}^{T}D_{KL}\big(q_\phi(z_t|\mathbf{x}_{\leq t}, y_t), p_\theta(z_t)\big) + \sum_{t=1}^{T}\alpha \log q_\phi(y_t|\mathbf{x}_{\leq t}) \tag{A2}$$

where $\alpha$ controls the relative weight between generative and purely discriminative learning.

### A.3. Model M3

Here we derive the objective function in Eq. 6. The generative and recognition models are factorized as:

$$p_\theta(X_{\leq T}, z_{\leq T}|\mathbf{x}_{\leq T}) = \prod_{t=1}^{T}p_\theta(X_t|z_{\leq t}, \mathbf{x}_{\leq t})p_\theta(z_t)$$

$$q_\phi(z_{\leq T}|\mathbf{x}_{\leq T}) = \prod_{t=1}^{T}q_\phi(z_t|\mathbf{x}_{\leq t})$$

---

**Algorithm 2:** $PatComClassM1(x_{1:\tau}^{(1:2)})$

---

**Recognition Model**

1 **for** $i \leftarrow 1$ *to* 2 **do**

2     $h_\tau^{enc_i} \leftarrow RNN_\phi^{enc}(x_{1:\tau}^{(i)}, h_{\tau-1}^{enc_i})$

3     $[\mu_\tau^{(i)} \, ; \Sigma_\tau^{(i)}] \leftarrow \varphi^{enc}(h_\tau^{enc_i})$

**end**

**Product of Experts**

4 $z_\tau \sim \mathcal{N}(\mu_\tau, \Sigma_\tau)$, where $\Sigma_\tau \leftarrow \Big( \sum_{i=1}^{2} \Sigma_\tau^{(i)^{-2}} \Big)^{-1}$, $\mu_\tau \leftarrow \Big( \sum_{i=1}^{2} \mu_\tau^{(i)} \Sigma_\tau^{(i)^{-2}} \Big) \Sigma_\tau$

**Generative Model**

Pattern completion

5 $h_\tau^{dec_1} \leftarrow RNN_\theta^{dec}(z_\tau, h_{\tau-1}^{dec_1})$

6 $\hat{X}_\tau \leftarrow f_\sigma(h_\tau^{dec_1}, \hat{X}_{\tau-1})$

Classification Model

7 $h_\tau^{dec_2} \leftarrow RNN_\theta^{dec}(z_\tau, h_{\tau-1}^{dec_2})$

8 $\hat{y}_\tau \leftarrow softmax(h_\tau^{dec_2})$

---

The variational lower bound (ELBO) on the log-likelihood of the generated data, $\log p_\theta(X_{\leq T}|\mathbf{x}_{\leq T})$, is derived as:

$$\mathbb{E}_{q_\phi(z_{\leq T}|\mathbf{x}_{\leq T})} \Big[ \log p_\theta(X_{\leq T}|\mathbf{x}_{\leq T}) \frac{q_\phi(z_{\leq T}|\mathbf{x}_{\leq T})}{q_\phi(z_{\leq T}|\mathbf{x}_{\leq T})} \Big]$$

$$= \mathbb{E}_{q_\phi(z_{\leq T}|\mathbf{x}_{\leq T})} \Big[ \log \frac{p_\theta(X_{\leq T}, z_{\leq T}|\mathbf{x}_{\leq T})}{p_\theta(z_{\leq T}|\mathbf{x}_{\leq T})} \frac{q_\phi(z_{\leq T}|\mathbf{x}_{\leq T})}{q_\phi(z_{\leq T}|\mathbf{x}_{\leq T})} \Big]$$

$$= \mathbb{E}_{q_\phi(z_{\leq T}|\mathbf{x}_{\leq T})} \Big[ \sum_{t=1}^{T} \log \frac{p_\theta(X_t|z_{\leq t}, \mathbf{x}_{\leq t}) p_\theta(z_t)}{p_\theta(z_t|\mathbf{x}_{\leq t})} \frac{q_\phi(z_t|\mathbf{x}_{\leq t})}{q_\phi(z_t|\mathbf{x}_{\leq t})} \Big]$$

$$= \mathbb{E}_{q_\phi(z_{\leq T}|\mathbf{x}_{\leq T})} \Big[ \sum_{t=1}^{T} \Big[ \log p_\theta(X_t|z_{\leq t}, \mathbf{x}_{\leq t}) - \log \frac{q_\phi(z_t|\mathbf{x}_{\leq t})}{p_\theta(z_t)} + \log \frac{q_\phi(z_t|\mathbf{x}_{\leq t})}{p_\theta(z_t|\mathbf{x}_{\leq t})} \Big] \Big]$$

$$\geq \mathbb{E}_{q_\phi(z_{\leq T}|\mathbf{x}_{\leq T})} \Big[ \sum_{t=1}^{T} \log p_\theta(X_t|z_{\leq t}, \mathbf{x}_{\leq t}) \Big] - \sum_{t=1}^{T} D_{KL}\big(q_\phi(z_t|\mathbf{x}_{\leq t}), p_\theta(z_t)\big)$$

After adding the classification loss, the final objective function can be written as:

$$\mathbb{E}_{q_\phi(z_{\leq T}|\mathbf{x}_{\leq T})} \Big[ \sum_{t=1}^{T} \log p_\theta(X_t|z_{\leq t}, \mathbf{x}_{\leq t}) \Big] - \sum_{t=1}^{T} D_{KL}\big(q_\phi(z_t|\mathbf{x}_{\leq t}), p_\theta(z_t)\big) + \log q_\pi(y|X) \quad \text{(A3)}$$

where $q_\pi(y|X)$ is the classification model whose input is the entire image (completed pattern) and not a sequence of observations. So the subscript $t$ is dropped.

---

**Algorithm 3:** $PatComClassM2(x_{1:\tau}^{(1:2)}, y_{1:\tau})$

---

**Classification Model**

1  $h_\tau^{cls} = RNN_\alpha^{cls}(h_{\tau-1}^{cls}, \mathbf{x}_{1:\tau})$

2  $\hat{y}_\tau = softmax(h_\tau^{cls})$

**Recognition Model**

3  **for** $i \leftarrow 1$ *to* 2 **do**

4  $\quad h_\tau^{enc_i} \leftarrow RNN_\phi^{enc}(x_{1:\tau}^{(i)}, h_{\tau-1}^{enc_i})$

5  $\quad [\mu_\tau^{(i)} ; \Sigma_\tau^{(i)}] \leftarrow \varphi^{enc}(h_\tau^{enc_i})$

**end**

6  **if** *labels are present* **then**

7  $\quad h_\tau^{enc_3} \leftarrow tanh(y_\tau)$

**else**

8  $\quad h_\tau^{enc_3} \leftarrow tanh(\hat{y}_\tau)$

9  $[\mu_\tau^{(3)} ; \Sigma_\tau^{(3)}] \leftarrow \varphi^{enc}(h_\tau^{enc_3})$

**Product of Experts**

10  $z_\tau \sim \mathcal{N}(\mu_\tau, \Sigma_\tau)$, where $\Sigma_\tau \leftarrow \Big( \sum_{i=1}^{3} \Sigma_\tau^{(i)^{-2}} \Big)^{-1}$, $\mu_\tau \leftarrow \Big( \sum_{i=1}^{3} \mu_\tau^{(i)} \Sigma_\tau^{(i)^{-2}} \Big) \Sigma_\tau$

**Generative Model**

Pattern Completion

11  $h_\tau^{dec(1)} \leftarrow RNN_\theta^{dec}(z_\tau, h_{\tau-1}^{dec_1})$

12  $\hat{X}_\tau \leftarrow f_\sigma(h_\tau^{dec_1}, \hat{X}_{\tau-1})$

---

## Appendix B. Visualization of fixation maps

Visualization of the fixation maps obtained from our model (M1), RAM (Mnih et al., 2014), and the participants in (Baruah et al., 2023b), on uppercase and lowercase alphabets are shown in Figs. A1 and A2 respectively.

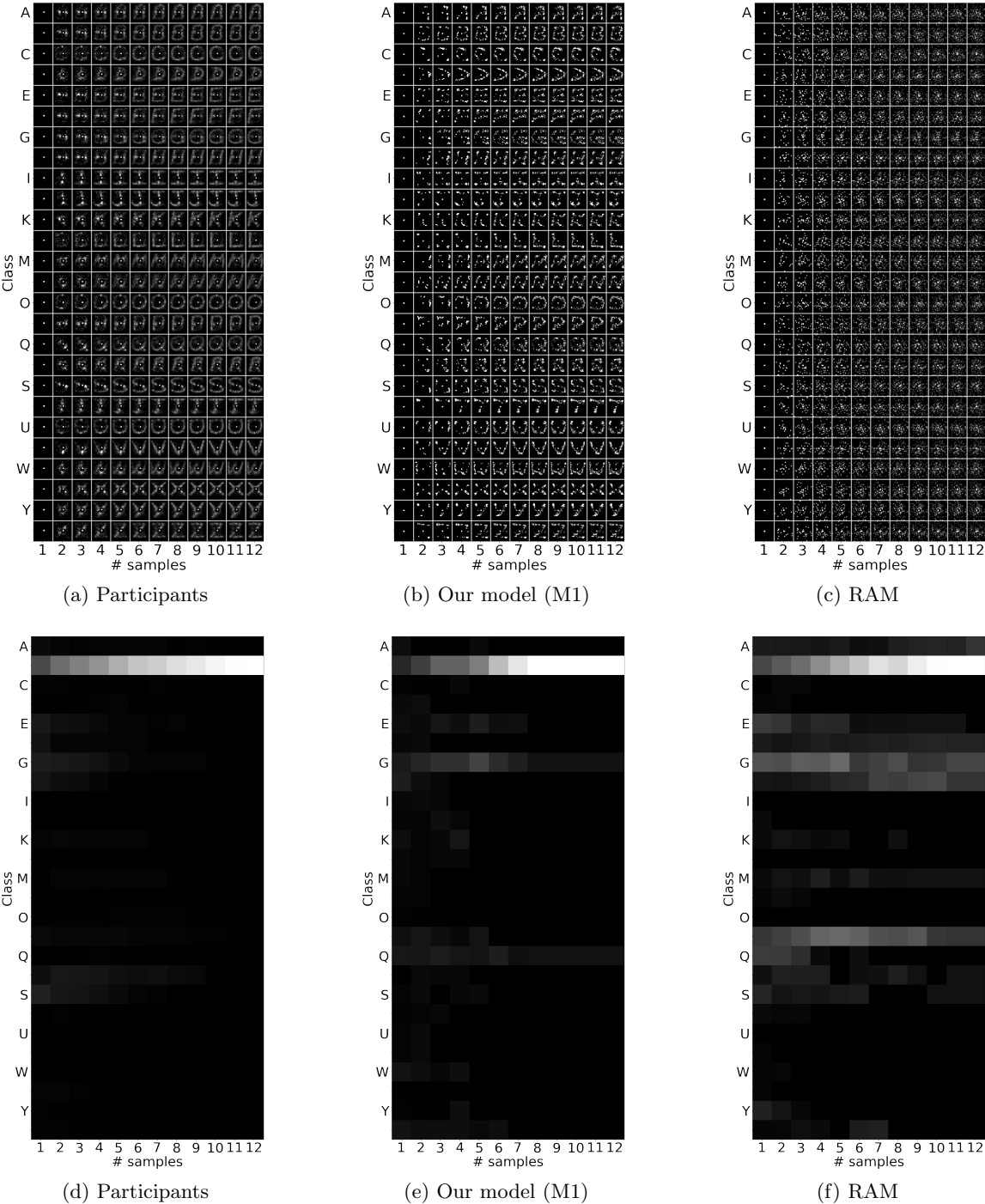

Figure A1: (a)–(c) Distribution of sampling locations (or fixation maps) for each uppercase alphabet and each sampling instant. Qualitatively, the participants' fixation maps are more similar to our model's than RAM's. (d)–(f) Class distribution for class 'B'. The distributions are obtained by averaging the responses over all stimuli presented from each class. Each row corresponds to a class, and each column corresponds to a sampling instant which increases from left to right.

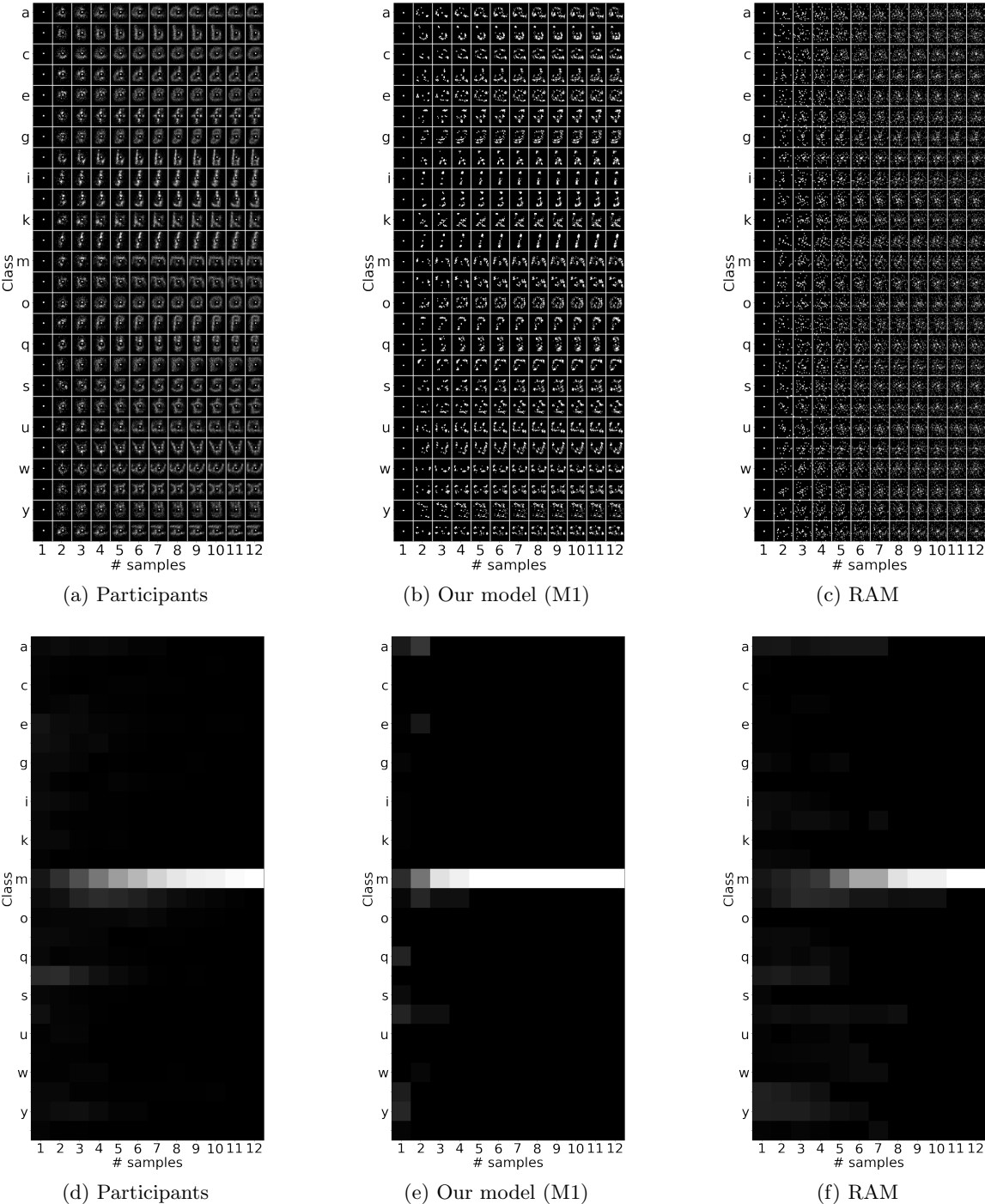

Figure A2: (a)–(c) Distribution of sampling locations (or fixation maps) for each lowercase alphabet and each sampling instant. Qualitatively, the participants' fixation maps are more similar to our model's than RAM's. (d)–(f) Class distribution for class 'm'. The distributions are obtained by averaging the responses over all stimuli presented from each class. Each row corresponds to a class, and each column corresponds to a sampling instant which increases from left to right.

