# OpenReview forum: "An Attention-based Predictive Agent for Handwritten Numeral/Alphabet Recognition via Generation"
_NeurIPS.cc/2023/Workshop/Gaze_Meets_ML — Gaze Meets ML 2023 Oral_

### Official Review · Reviewer_P3TF · 2023-10-18
**Overall, the paper addresses an interesting topic of attention-based agent model that learns to classify handwritten numerals/alphabets from images by generating them.**

**Rating:** 9
**Confidence:** 4

**Review:**

Peer-Review Report
Review of paper titled "An Attention-based Predictive Agent for Handwritten Numeral/Alphabet Recognition via Generation”

 This study addressed the predictive agent model that actively samples its visual environment through a sequence of glimpses. At each sampling instant, the authors use the  model to implement a perception-action loop to optimize an objective function. The action (attention) is modeled as proprioception in a multimodal setting and is guided by perceptual prediction error, not by reinforcement. The authors evaluated three variants of this model for handwriting generation and recognition on images of handwritten numerals and alphabets from benchmark datasets. The authors were able to answer  where and what to sample by jointly minimizing the classification and generation errors. The authors were able to verify their claim that their proposed model is more efficient in handwritten numeral/alphabet recognition than human participants in a recently published study as well as a highly cited attention-based reinforcement model.
The authors model proves to be the first known attention-based agent to interact with and learn end-to-end from images for recognition via generation, with high degree of accuracy and efficiency signifying the study novelty and new insight in the sense that the pattern completion function maps the partial sequences of perceptual and proprioceptive observations to the class label and completed perceptual pattern.
Overall, the paper addresses an interesting topic of attention-based agent model that learns
to classify handwritten numerals/alphabets from images by generating them.

The paper is well written and explained clearly, albeit there exist a few Questions:
Question 1: In page 4, Model 1 M1,  The model is trained end-to-end as well as model 2 M2 and model 3 M3 is trained differently. Then why was (in page 11)The classification accuracy from M1 is differs much with M2 but then very close M3, even though M1 and M2 are modelled similarly and M2 differently?
Question 2: In M3, does the classification modality shares parameters with the generation modality and does the generation modality shares parameters with the classification modality
Question 3:Oone more variant of the model in which the generative model is trained was included as M4 in almost all the tables and evaluations, however, these model M4 was not included in the case study in section two, why?
Overall flow of the manuscript is very good.
Recommendation: Accept

---

### Official Review · Reviewer_ampP · 2023-10-23
**The paper describes first attention based end-to-end agent for recognition via generation that beats human participants in efficiency for handwritten numeral/alphabet recognition.  The images are sampled as a sequence of observations and modeled as a pattern completion and classification problem.**

**Rating:** 9
**Confidence:** 3

**Review:**

**Originality :** The paper introduces first of it's kind attention based efficient agent that performs simultaneous generation and classification of handwritten alphabets. While a couple of models in the past have reported end-to-end generation and classification of handwritten numerals, only classification or generation accuracy (but not both) is reported for both of them. This paper differs from previous literature in that it reports accuracy on both generation and classification tasks together while also sampling the image as a sequence of observations and not in its entirety, unlike previous models did.

**Quality :**  The proposed model and its three variants are compared for efficiency and fixation maps with a popular reinforced model, recurrent attention model (RAM). Benchmarking experiments on a variety of datasets (MNIST, EMNIST and AttentionMNIST) demonstrate that the fixation maps generated by the proposed model are similar to those generated by the participants. The proposed model is also more data efficient when predicting a class compared to the participants and the RAM model.

**Clarity :** The paper is very well written, background and related work has been cited well. The proposed model and its variations are easy to understand. The comparisons are well drawn and the terms/metrics used throughout the paper are well explained.

**Significance :** Attention based agents that observe the visual environment and learn what action to take by minimizing sensory prediction error in a closed loop system can be very valuable in day to day tasks such as deciphering handwritten alphabets. The paper poses one of its kind end to end attention agent that is more efficient and performs better than human participants.

**Concern :** The human population comparison data uses mcAT dataset, recorded from 382 distinct participants presented with images selected from benchmark datasets. Since this dataset involved participants to look at the static images under free-viewing conditions and with ample time at hand, the comparisons of the proposed model w.r.t the human population for efficiency might have some room for improvement.

---

### Official Review · Reviewer_xBfA · 2023-10-24
**Well written paper and very well designed study with extensive evaluation techniques**

**Rating:** 9
**Confidence:** 4

**Review:**

The authors propose a model that utilizes the perception-action loop and creates a model for numbers and letter recognition based on an agent framework. This is very well aligned with the research in the eye tracking community where the scan path of different users in order to perform a certain task is investigated.

The use of glimpse as a mechanism representing loosely our foveation is an added strength of this paper very well applicable to how our information integration leads to the perception of a scene.

It would be very helpful for the readers if the authors clarify whether the order of fixations i.e. if two exactly similar fixation map, but one having a completely different order would be picked up in the three metrics that was proposed or not. If not, what would this inform us? As in, would the proposed model begin with attending to a certain part of the scene always if it was asked to repeat the same digit/letter over and over? What would the role of order of glimpse tell us about the behavior of the model?

---

### Meta-Review · Area_Chair_7HZf · 2023-10-26

**Recommendation:** Accept (Oral)
**Confidence:** 4

**Metareview:**

The paper is the first known attention-based agent to interact end-to-end with images for recognition via generation. It achieves a high degree of accuracy and efficiency, demonstrating this study's novelty and new insights. The pattern completion function maps partial sequences of perceptual and proprioceptive observations to the class label and completed perceptual pattern.

Overall, the paper addresses an interesting topic of an attention-based agent model that learns to classify handwritten numerals and alphabets from images by generating them. All authors agreed that the simultaneous generation and classification of handwritten alphabets using agent-based methods was an exciting prospect. They recognized the benefits of this approach and benefits over the existing popular Recurrent Attention Model (RAM) approach. The authors also acknowledged the paper's clear writing, well-designed experiments, ablation studies, and comparisons with existing methods. They had minor concerns about the study design choices and clarification of models M1, M2, and M3 in the paper.

---

### Decision · Program_Chairs · 2023-10-26

Accept (Oral)